

# Observed and simulated full-depth ocean heat content changes for 1970-2005

Lijing Cheng[1], Kevin E. Trenberth[2], Matthew D. Palmer[3], Jiang Zhu[1], John P. Abraham[4]

[1]International Center for Climate and Environment Sciences, Institute of Atmospheric Physics, Chinese Academy
of Sciences, 100029, Beijing, China
[2]National Center for Atmospheric Research, Boulder, Colorado
[3] Met Office Hadley Centre, FitzRoy Road, Exeter, EX1 3PB, UK
[4] University of St. Thomas, St. Paul, MN, USA

*Correspondence to*: John Abraham (jpabraham@stthomas.edu); Lijing Cheng (chenglij@mail.iap.ac.cn)

**Abstract.** Greenhouse-gas emissions have created a planetary energy imbalance that is primarily manifested by increasing ocean heat content (OHC). Updated observational estimates of full-depth OHC change since 1970 are presented that account for recent advancements in reducing observation errors and biases. The full-depth OHC has increased by 0.74 [0.68, 0.80] $\times 10^{22}$ J/yr (0.46 Wm$^{-2}$) and 1.22 [1.16-1.29] $\times 10^{22}$ J/yr (0.75 Wm$^{-2}$) for 1970-2005 and 1992-2005 respectively, with 5% to 95% confidence interval of the median. The CMIP5 models show large spread in OHC changes, but the ensemble median has excellent agreement with our observational estimate: 0.68 [0.54-0.82] $\times 10^{22}$ J/yr (0.41 Wm$^{-2}$) from 1970 to 2005 and 1.25 [1.10-1.41] $\times 10^{22}$ J/yr (0.77 Wm$^{-2}$) from 1992 to 2005. These results increase confidence in both the observational and model estimates to quantify and study changes in Earth's energy imbalance over the historical period. We suggest that OHC be a fundamental metric for climate model validation and evaluation.

## 1 Introduction

Since the beginning of the industrial revolution, increased emissions of long-lived greenhouse gases such as carbon dioxide have resulted in an accumulation of thermal energy in the climate system (*Trenberth et al.*, 2014; *von Schuckmann et al.*, 2016) via the associated net energy imbalance at Earth's top-of-atmosphere (TOA). It is estimated that more than 90% of the excess heat is stored in the ocean and is manifested by ocean warming (*Loeb et al.*, 2012; *Balmaseda et al.*, 2013; *Rhein et al.*, 2013; *Trenberth et al.*, 2014), i.e. an increase of global ocean heat content (OHC) (*Lyman et al.*, 2010; *Levitus et al.*, 2012; *Abraham et al.*, 2013). Due to the ocean's dominant role in the global energy storage changes, the rate of OHC change provides a strong constraint on Earth's energy imbalance on interannual and longer timescales (*Palmer and McNeall*, 2014; *Trenberth*, 2015). Numerous efforts





have been made to detect the historical OHC change (*Smith and Murphy*, 2007; *Domingues et al.*, 2008; *Palmer and Haines*, 2009; *Ishii and Kimoto*, 2009; *Lyman et al.*, 2010; *Levitus et al.*, 2012; *Balmaseda et al.*, 2013; *Cheng et al.*, 2015a) and attribute causes to its variation (*Palmer et al.*, 2009; *Gleckler et al.*, 2012). However, large uncertainties exist in OHC estimates (*Abraham et al.*, 2013; *Balmaseda et al.*, 2013; *Rhein et al.*, 2013), which

can confound our understanding of the changes in Earth's energy imbalance since the 1970s.

A major source of error in the historical *in situ* temperature data that underpin OHC estimates are time-varying systematic biases in expendable bathythermograph (XBT) temperature measurements (*Gouretski and Koltermann*, 2007; *Lyman et al.*, 2010; *Abraham et al.*, 2013). Numerous correction schemes have been proposed

to remove the time-varying XBT biases (*Cheng et al.*, 2015b), but these schemes vary in their formulation and performance. Hence, the XBT community met in 2014 and made a series of recommendations on the factors that should be accounted for when designing and implementing an XBT bias-correction scheme (*Cheng et al.*, 2015b). Only one bias correction scheme (*Cheng et al.*, 2014) meets all of these recommendations and has been shown to correct the overall bias to less than 0.02°C (for the 0-700m layer, less than 10% of the total 0-700m temperature

change since 1970), and also reduce the spatio-temporal variation of bias.

Prior to 2004, observations of the upper ocean were predominantly confined to the Northern Hemisphere and concentrated along major shipping routes; the Southern Hemisphere is particularly poorly observed. In this century, the advent of the Argo array of autonomous profiling floats (*Roemmich et al.*, 2015; *von Schuckmann et*

*al.*, 2014) has significantly increased ocean sampling to achieve near-global coverage for the first time over the upper 1800m since about 2005.

The lack of historical data coverage requires a gap-filling (or mapping) strategy to infill the data gaps in order to estimate the global integral of OHC. A pioneering study showed that an improved strategy for gaps-filling method

and corrections for XBT biases improved the consistency between models and observations of upper 700m OHC (*Domingues et al.*, 2008). Owing to sparse observations in the Southern Hemisphere, *Durack et al.*, (2014) explored this region as a primary source of under-estimation of OHC trends using climate models from the Coupled Model Inter-comparison Project Phase 3/5 (CMIP 3/5) (*Meehl et al.*, 2007; *Taylor et al.*, 2012). *Cheng and Zhu*, (2014) examined the observation system evolution in this century, identifying a spurious signal from

2001-2003 in global OHC estimates due to inadequate sampling of the Southern Hemisphere prior to Argo. Accordingly, these studies imply that many past estimates likely underestimate the long-term trend.



The aim of this study is to use these improved XBT bias corrections and gap-filling methods designed to minimize the impact of historical sampling changes and to confront CMIP5 models with the state-of-the-art observational estimates of OHC change. We note that the work presented here is broadly similar to the recent study of Gleckler

et al (2016) and provides an important independent verification of some of their key findings. However, the present study also makes use of a larger number of CMIP5 models (24 compared to 15) and observation-based estimates of the 0-700m ocean heat content changes (8 compared to 3), including improved XBT bias corrections and new mapping approaches. We are therefore able to more fully characterise the uncertainties associated with CMIP5 models and place our new observation-based estimates of OHC in the context of several previous estimates

(including those of Gleckler et al). The manuscript is arranged as follows. In section 2 the data and methods are introduced. The various observation-based OHC estimates used are discussed in Section 3.1, CMIP5 model similatuions presented in Section 3.2. We summarize our findings in section 4.

### 2 Data and Methods

The new observation-based estimates of OHC presented here use the XBT bias correction scheme from *Cheng et*

*al.*, (2014) applied to the most recent version of the World Ocean Database (WOD2013) (*Boyer et al.*, 2013). Because the choice of reference climatology to compute anomalies can lead to errors due to the sparseness and inhomogeneity of the historical ocean sampling (*Lyman and Johnson*, 2014; *Cheng and Zhu*, 2015) it is preferable to construct the climatology based on data with near-global data coverage (*Cheng and Zhu*, 2015), i.e., the Argo period. In this study, we use a climatology constructed for the period 2008-2012, similar to *Cheng and Zhu*, (2014)

and *Cheng et al.*, (2015a).

We apply two approaches to mapping the OHC data. The first (*Cheng and Zhu*, 2014) (hereafter termed the CZ14 method) calculated annual mean OHC in data-rich areas (defined as Ship Area) and a linear OHC trend in data-sparse regions (defined as Argo-Ship Area). Then the two estimates are summed to get the global OHC. The

second approach is an extension to CZ14 that uses flexible grid sizes to retain greater spatial information while ensuring an adequate number of observations in each grid box. OHC in each 1º by 1º grid in poorly sampled regions (Argo-Ship Area defined in CZ14) is calculated by averaging OHCs over a large latitude-longitude grid with sizes of 5º by 5º, 5º by 10º, 5º by 20º, 1º by 40º, 8º by 40º, and 10º by 40º separately to ensure that all regions have data coverage (Figure 1). The gridded averaged anomalies are then integrated to get global OHC. This

method (Gridded method hereafter) maintains the observed OHC in data-rich regions without smoothing and



provides a smooth OHC field in data-sparse regions. This is appropriate for the Southern Hemisphere where there is more homogeneity, less land and an absence of boundary currents.

In addition to our new observation-based OHC estimates, we also present two recent sets of estimates that make
use of dynamical models. The first uses climate model simulations (*Durack et al.*, 2014) to adjust five of the existing upper 700m OHC estimates (*Domingues et al.*, 2008; *Durack and Wijffels*, 2010; *Ishii and Kimoto*, 2009; *Levitus et al.*, 2012; *Smith and Murphy*, 2007), which may have underestimated trends due to the very limited data coverage in the Southern Hemisphere. In addition to the *Durack et al.*, (2014) global OHC adjustments which are based on comparing hemispheric ratios of heat uptake in the CMIP5 models, it is desirable to also use other
estimates from independent studies. The second is the ORAS4 dataset, which is an ocean reanalysis product (*Balmaseda et al.*, 2013). Ocean reanalyses have the advantage of synthesizing a large number of different observations into a dynamically consistent estimate of the historical ocean state and can potentially provide greater physical insight into the mechanisms of OHC change (*Balmaseda et al.*, 2013; *Palmer et al.*, 2015; *Xue et al.*, 2012). The five ensemble members in ORAS4 sample plausible uncertainties in the wind forcing, observation
coverage, and the deep ocean. OHCs with the layers of 0-700m, 700-2000m and 2000-bottom are all used in this study.

Combining our new OHC estimates with  existing estimates provides an ensemble of observational-based estimates of historical upper 0-700m OHC changes and the spread is a simple measure of the observational
uncertainty. Differences across the ensemble arise not only from mapping methods, but also from choice of climatology, input data quality control procedures and XBT correction scheme (*Palmer et al.,* 2010).

To arrive at estimates of full-depth OHC change, we adapted and adjusted the *Levitus et al.* (2012) estimate for the 700-2000m layer and for the deeper ocean for the period 1990-2010, we use information from *Purkey and*
*Johnson* (2010), which was also used in the IPCC-AR5 report (*Rhein et al,* 2013). Prior to 1990, there is a larger uncertainty regarding the rate of deep-ocean warming during 1970-1991 period. Because the upper 700-2000m oceans show an approximate tripling of the heating rate from 1992-2005 compared to 1970-1991 (as shown in Fig. 2, green curve), we assume a proportionate increase in heat uptake in the deep ocean (2000m-bottom). For uncertainty calculations, we use a lower bound of no deep-ocean warming prior to 1992 and an upper bound of
an unchanging linear trend from 1970-2005, as assumed in *Church et al.*, (2011). Because this is an important assumption, it is valuable to assess the uncertainties involved. We show that the difference of this lower and upper



bound of the 700m-bottom OHC change is equal to ~13% (~10%) of the full-depth OHC change during 1970-1991 (1970-2005), which indicates the maximum error induced by this assumption. The ORAS4 data also provide estimates on OHC changes deeper than 700m. We estimate the uncertainty for the OHC changes below 700m by computing the standard error from the ensemble members of *Levitus et al.*, (2012), *Purkey and Johnson* (2010)

and ORAS4 ensembles and presenting the 5-95% confidence interval.

We compare our observation-based OHC ensemble with 24 CMIP5 model simulations (Table 1) of historical OHC changes. Climate models suffer from so-called "drift" (*Sen Gupta et al.*, 2013; *Hobbs et al.*, 2015), i.e. spurious long-term trends arising the slow model adjustment to the initial conditions and/or imperfect

representation of the energy budget. This drift can bias the long-term representation of the ocean temperature, especially in deeper layers. Because there is no general consensus on how to correct for climate drift in models, we applied two different drift correction strategies by using available pre-industrial control ("piControl") runs of 24 CMIP5 models. We applied both a linear and a quadratic fit to the OHC time series of pi-control runs for OHC0-700m, 700-2000m and 2000-6000m. The resulting regression function is removed from the historical

simulations for each model. The two methods show nearly identical results (Table 2 and Table 3) and we present the results for quadratic drift correction as the basis of our discussions.

To quantify the OHC changes for a given time period, we fit a linear trend. An alternative method calculating the OHC difference between the two ends of a time series shows consistent results (compare Table 2 with Table 3).

For both observational-based OHC and CMIP5-OHC results, we calculate the median of the ensemble to reduce the impact of outliers, together with the 5% to 95% confidence interval of the median assuming that the values were independently and randomly sampled from a population distributed according to a Gaussian distribution. Therefore, the 5%-95% confidence interval is: ± Standard Error × 2.10. The *Student-t* test is used to examine the significance of the difference between observations and CMIP5 models.

**3 Results**

**3.1 Observation-based full-depth OHC estimates**

Figure 2 presents the observation-based 0-700m OHC estimates by using the methods listed in the previous section, after taking the Southern Hemisphere sampling bias into account. The updated 0-700m OHC estimate based on CZ14 method indicates a total upper ocean warming of approximately $21.0 \times 10^{22}$ J, equal to a linear trend of

$0.56 \times 10^{22}$ J/yr (or 0.35 Wm$^{-2}$, averaged over the global surface area) from 1970 to 2005. The six individual





Gridded Method estimates (based on six choices of grid size) (Figure 2) span a range of 0.52-0.58×10$^{22}$J/yr during the 1970-2005 period, consistent with the CZ14 estimate. In addition, according to *Durack et al.*, (2014), the change in global 0-700m OHC over the period 1970-2005 increased by 0.43-0.56×10$^{22}$ J/yr (Figure 2). One estimate (*Smith and Murphy*, 2007), which shows much smaller values than the others, is discounted. ORAS4

reanalyses shows a range of 0.49-0.53×10$^{22}$ J/yr for the 0-700m OHC.

The collection of the different observational OHC estimates discussed above (16 individual estimates) provides current best estimates of OHC and also indicates the uncertainties (Figure 3a). Although all OHC estimates are based on an essentially the same temperature profile database, they use four different methods, and hence their

differences give an indication of the uncertainty. The total OHC change of the upper 700m layer has increased by 0.55×10$^{22}$ J/yr (0.34 Wm$^{-2}$), which is the median among all of the ensemble members, with 5%-95% confidence interval of 0.50-0.60×10$^{22}$J/yr.

On the other hand, it is worthwhile noting that the comparison of CZ14, Gridded Method and ORAS4 results

show inconsistency of OHC changes on interannual time-scales (Figure 2), indicating that the errors in OHC estimates are still larger than the inter-annual variability, as shown in *Abraham et al.* (2013). However, all of the estimates show the OHC decreases after the major volcano eruptions: El Chichón in March–April 1982 and Pinatubo in June 1991 (Figure 2). The OHC change after the two volcano eruptions is approximately assessed by subtracting the OHC one year before the eruption from the OHC in the second year after eruption. It shows a 0-

700m OHC decrease of ~ −2.67 [−3.28, −2.06]×10$^{22}$ J after El Chichón and ~ −2.72 [−3.97, −1.47] ×10$^{22}$ J after Pinatubo, indicating the strong ocean cooling. The negative radiative forcing to the ocean (and climate system) due to the volcano eruption is probably the major reason for this decrease (*Church et al.*, 2005, *Domingues et al.*, 2008; Balmaseda et al. 2013). But the unforced ocean variability (such as ENSO) and the insufficiency of data coverage (which could induce spurious inter-annual OHC change) could partly contribute to the values calculated

above. There is also indication of substantial heat discharge from the upper 700m ocean following the extreme 1997-1998 El Niño event (*Balmaseda et al.*, 2013; *Roemmich and Gilson*, 2011) with CZ14 estimate showing a lesser response than the other estimates partly due to their assumption of a linear long-term change in the data-sparse region. This 0-700m OHC decrease is ~ −2.73 [−3.27, −2.20] ×10$^{22}$ J after the 1997–1998 El Niño averaging over the all products. The decrease is calculated by the difference of OHC between 2000 and 1998 for

ORAS4 and between 2000 and 1999 for CZ14 and Gridded Method, since the latter products appear a delayed response. The differences among the datasets indicate the uncertainties of both gap-filling methods and the



processes of OHC redistribution during ENSO represented by re-analyses (ORAS4) in the vertical in Pacific Ocean and into the other ocean basins via atmosphere teleconnections (*Mayer et al.*, 2013).

For deeper ocean layers, we adopt the 700-2000m ocean heat content estimate from 1970 to 2005 in *Levitus et al.*,
(2012), where all of the historical *in situ* data are objectively analyzed. According to *Levitus et al.*, (2012), the 700-2000 m ocean warmed by $0.12\times10^{22}$ ($0.17\times10^{22}$) J/yr or 0.075 (0.106) Wm$^{-2}$ over global surface since 1970 (1992). For the abyssal (2000m-bottom) OHC changes, according to the strategies provided in the Methods section, we estimate a deep ocean warming of $0.025\times10^{22}$ ($0$-$0.075\times10^{22}$) J/yr or 0.016 (0-0.046) Wm$^{-2}$ during the 1970-1991 period and $0.12\times10^{22}$ J/yr (0.075 Wm$^{-2}$) during 1992-2005. According to the two estimates at two layers, the
ocean warming rate deeper than 700m is $0.145\times10^{22}$ J/yr (0.090 Wm$^{-2}$) during 1970-2005. However, as we discussed above, the traditional method from *Levitus et al.*, (2012) is likely to underestimate the long-term trend, and this is also the case for 700-2000m estimate on OHC change. Hence it is also valuable to use ORAS4, which provides alternative estimates of 700-2000m/2000m-bottom OHC changes and also provide an assessment of the uncertainty. It is shown from the recent Reanalyses Intercomparison Project (*Palmer et al.*, 2015) that there remain
large biases in the deeper ocean, because there is limited data available 700m (historical), and hence it is a challenge for assimilation to deliver information to the model in those layers. ORAS4 shows the deeper 700m-bottom ocean warming of $0.09{\sim}0.24\times10^{22}$ J/yr (0.056~0.150 Wm$^{-2}$) since 1970, indicating large uncertainties but generally consistent with the previous assessments based on *Levitus et al.* (2012) and *Purkey and Johnson* (2010).

By summing OHCs for the different layers 0-700m, 700-2000m and 2000m-bottom, the observation-based full-depth OHCs are obtained. All of these results (Figure 3b) indicate a range of full-depth ocean warming of 0.50-$0.79\times10^{22}$J/yr (0.31-0.50 Wm$^{-2}$) over the 36-year period (1970-2005, again calculated by linear trend). The median of the different estimates is 0.74 [0.68, 0.80] $\times10^{22}$ J/yr (1.22 [1.16-1.29]$\times10^{22}$ J/yr) since 1970 (since 1992) with the values in brackets representing the 5% and 95% confidence intervals of the median. This is equivalent to a
global energy imbalance of 0.46 [0.42, 0.50] Wm$^{-2}$ (0.75 [0.69, 0.81] Wm$^{-2}$) averaged over Earth's surface area since 1970 (1992). Furthermore, after the two major volcano eruptions, the total OHC decrease is ~ $-2.42$ [$-3.28$, $-1.56$]$\times10^{22}$ J for El Chichón and ~$-3.19$ [$-4.92$, $-1.67$] $\times10^{22}$ J for Pinatubo. Following the major 1997-1998 El Niño event, the total OHC decreases by ~ $-1.85$ [$-2.62$, $-1.10$] $\times10^{22}$ J. This indicates a substantial rearrangement of heat from 0-700m to deeper ocean, since most ensemble members show smaller full-depth heat loss than for
the 0-700m layer.





### 3.2 Climate Model Assessments

It is important to quantify the agreement of models, such as those in CMIP5 (*Taylor et al.*, 2012; *Durack et al.*, 2014; *Gleckler et al.*, 2016), with observations both to validate the models and also reconcile the observations with expectations based on radiative forcing estimates. Comparisons are made (Fig. 3a) between the updated OHC

observations and 24-run ensemble climate models from 1970 to 2005, which is the limit for reasonable observational coverage (*Lyman and Johnson*, 2014) and is also restricted to the end time of the CMIP5 model runs for historical simulations (2005).

The distribution of OHC0-700m from the 24 models after a correction of "climate drift" (see Methods), shows an

ensemble median of 0.42 [0.32-0.51] $\times 10^{22}$ J/yr (0.26 [0.19-0.37] Wm$^{-2}$) for the 1970-2005 time period and 0.89 [0.77-1.02] $\times 10^{22}$ J/yr (0.55 [0.48-0.64] Wm$^{-2}$) for 1992-2005. The sensitivity of the results to the climate drift correction is very small (within 0.03 $\times 10^{22}$ J/yr) when two different climate drift correction methods are applied (as shown in Tables 2, 3 and Figure 4). For the 1970-2005 period, the median of the CMIP5 models is significantly smaller than observations (0.55 [0.50-0.60]$\times 10^{22}$ J/yr), indicating that the models under-estimate the upper 700m

OHC change since 1970. But within the 1992-2005 period, the median of the CMIP5 models falls into the confidence interval of the existing observational estimates, indicating that the ensemble median of models agree very well with observational estimates in the recent period.

For full-depth OHC, drift-corrected CMIP5 models show the total OHC change by 0.68 [0.54-0.82]$\times 10^{22}$ J/yr

(0.42 [0.34-0.51] Wm$^{-2}$) from 1970-2005 and 1.25 [1.10-1.41]$\times 10^{22}$ J/yr (0.78 [0.68-0.88] Wm$^{-2}$) during 1992-2005 (Figure 3). The CMIP5 ensemble median again shows very good agreement with observations for both 1970-2005 (0.74$\times 10^{22}$ J/yr) and for 1992-2005 (1.22$\times 10^{22}$ J/yr). The central estimates of observation-based and CMIP5 OHC change are consistent within the estimated uncertainty. The total OHC decrease after the two major volcano eruption is ~ −0.60 [−0.81, −0.38]$\times 10^{22}$ J for El Chichón and ~−1.47 [−1.93, −1.00] $\times 10^{22}$ J for Pinatubo, which

are weaker than for observations.

Table 2 provides a summary of observed and simulated OHC change for different time periods and depths. CMIP5 results are shown for the upper ocean both with linear and quadratic drift corrections. Within the drift-corrected CMIP5 models, the rate of ocean warming has nearly doubled since 1992 (Figure 5, Table 2): 0.56 [0.43,

0.68]$\times 10^{22}$ J/yr within 1970-1991 (~0.35 [0.26, 0.43] Wm$^{-2}$ over global surface) compared to 1.25 [1.10, 1.41]$\times 10^{22}$ J/yr during 1992-2005 (~0.77 [0.67, 0.87] Wm$^{-2}$) for the both the drift-corrected CMIP5 ensembles,



while for observations the corresponding values are 0.61 [0.53, 0.69] ×10$^{22}$ J/yr within 1970-1991 (~0.38 [0.33, 0.43] Wm$^{-2}$), and 1.22 [1.16, 1.29] ×10$^{22}$ J/yr during 1992-2005 (~0.75 [0.71, 0.80] Wm$^{-2}$). This provides evidence for an acceleration of ocean warming due to the increasing radiative forcing from rising greenhouse gases and from the effects of volcanic eruptions near the intersection of those two time periods (*Myhre et al.*, 2013). This acceleration of ocean warming is also found by a recent study (*Gleckler et al.*, 2016).

Furthermore, the model ensemble median of full-depth OHC agrees well with observations, but significantly under-estimates the OHC change in the upper 700m (Figure 5b). Yet OHC changes for 700-6000m in the models is likely to over-estimate the warming rate prior to 1990. Together these are indicative that the models might be too diffusive and the vertical distribution of heat may not be correct, as suggested by previous studies (*Forest et al.*, 2008; *Kuhlbrodt and Gregory*, 2012).

Although the comparison between the observational and CMIP5 full-depth OHC results in an insignificant difference, CMIP5 models show a large spread (Figure 3, 4, 5), indicating that there are still large uncertainties in model simulations of Earth's energy budget. There are two groups of models: seven models calculate a much smaller upper 700m ocean warming of less than 0.3×10$^{22}$ J/yr over 1970-2005; the other group shows 0-700m ocean warming of 0.3-0.75×10$^{22}$ J/yr (Figure 3a). The first group also shows much smaller full-depth OHC increase of less than 0.35×10$^{22}$ J/yr than the second: 0.35-1.05×10$^{22}$ J/yr over 1970-2005 (Figures 3b). The second group shows better agreement with observational estimates. The models with smaller values should be treated with caution in future analyses. The reasons why the models have large divergence are still an actively studied issue. *Frolicher et al.*, (2015) discussed the large range of model results and attributed a contribution of this to the differences in indirect aerosols. Additionally, CMIP5 has been missing post 2000 volcanic eruptions in these simulations as discussed in Glecker et al (2016), but this effect is shown to small and less than 0.1 W m$^{-2}$ as indicated in *Trenberth et al.*, (2014).

**4 Summary**

This study presents new estimates of observed OHC change since 1970 based on improved mapping methods and XBT bias corrections. Our results suggest that previous IPCC-AR5 observational estimates of 0-700m OHC change of ~0.26 W m$^{-2}$ may be too low, typically by about ~25% compared to our findings here (~0.35 W m$^{-2}$), supporting the conclusions of *Durack et al.* (2014) based on somewhat different constraints. Our estimates of full-depth OHC change show remarkably good agreement with the CMIP5 ensemble median response during 1970-



2005 and gives us confidence that the climate models are not systematically biased in their simulation of historical variations in Earth's energy imbalance over this period.

The present work demonstrates how improvements in OHC estimation methods have led to a greater degree of consistency with climate model simulations of long-term changes in Earth's energy budget. In turn this allows an evaluation of the models and suggests that some may not be credible. Further work is needed to understand the spatial patterns of ocean heat uptake and TOA changes over the historical past as a means of assessing potential model deficiencies in key processes. Since 93% of the energy of global warming is stored in the ocean, our observational-based results indicate that the ocean component of the earth's heat imbalance of ~0.38 [0.33, 0.43] $Wm^{-2}$ from 1970 to 1991 and ~0.75 [0.71, 0.80] $Wm^{-2}$ from 1992 to 2005. With 0.07 $Wm^{-2}$ for the other components (*Trenberth et al*. 2014), the implied average energy imbalance after 1970 is 0.45 [0.40, 0.50] $Wm^{-2}$ and 0.82 [0.76, 0.88] $Wm^{-2}$ after 1992. For the period 1970-2005, our new value is about 15% larger than the central estimate of *Rhein et al.*, (2013) over the same period and could have important implications for closure of the sea level budget.

**Acknowledgments**

L.C. and J. Z. is supported by the project "Structures, Variability and Climatic Impacts of Ocean Circulation and Warm Pool in the Tropical Pacific Ocean" of the National Basic Research Program of China (2012CB417404), the Chinese Academy Sciences' Project "Western Pacific Ocean System: Structure, Dynamics and Consequences" (XDA11010405), and the National Natural Science Foundation of China (41506029). M.P. is supported by the Joint UK DECC/Defra Met Office Hadley Centre Climate Programme (GA01101) and his work represents a contribution to the Natural Environment Research Council DEEP-C project NE/K005480/1. KT is supported by DOE grant DE-SC0012711. NCAR is sponsored by the National Science Foundation. We thank NOAA/NODC, who made the observational ocean temperature dataset available. *We acknowledge the World* Climate Research Programme's Working Group on Coupled Modelling, which is responsible for CMIP, and we thank the climate modeling groups (listed in Table 1 of this paper) for producing and making available their model output. For CMIP the U.S. Department of Energy's Program for Climate Model Diagnosis and Intercomparison provides coordinating support and led development of software infrastructure in partnership with the Global Organization for Earth System Science Portals.



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



**Table 1. List of CMIP5 models and group names.**

| Modeling Center (or Group) | Institute ID | Model Name |
|---|---|---|
| Commonwealth Scientific and Industrial Research Organization (CSIRO) and Bureau of Meteorology (BOM), Australia | CSIRO-BOM | ACCESS1.0 |
| Beijing Climate Center, China Meteorological Administration | BCC | BCC-CSM1.1 BCC-CSM1.1(m) |
| Canadian Centre for Climate Modelling and Analysis | CCCMA | CanESM2 |
| National Center for Atmospheric Research | NCAR | CCSM4 |
| Community Earth System Model Contributors | NSF-DOE-NCAR | CESM1(FASTCHEM) |
| Commonwealth Scientific and Industrial Research Organization in collaboration with Queensland Climate Change Centre of Excellence | CSIRO-QCCCE | CSIRO-Mk3.6.0 |
| NOAA Geophysical Fluid Dynamics Laboratory | NOAA GFDL | GFDL-CM3 GFDL-ESM2G GFDL-ESM2M |
| NASA Goddard Institute for Space Studies | NASA GISS | GISS-E2-R |
| Met Office Hadley Centre (additional HadGEM2-ES realizations contributed by Instituto Nacional de Pesquisas Espaciais) | MOHC (additional realizations by INPE) | HadGEM2-CC HadGEM2-ES |
| Institut Pierre-Simon Laplace | IPSL | IPSL-CM5A-LR IPSL-CM5A-MR IPSL-CM5B-LR |
| Japan Agency for Marine-Earth Science and Technology, Atmosphere and Ocean Research Institute (The University of Tokyo), and National Institute for Environmental Studies | MIROC | MIROC-ESM |
| Atmosphere and Ocean Research Institute (The University of Tokyo), National Institute for Environmental Studies, and Japan Agency for Marine-Earth Science and Technology | MIROC | MIROC5 |
| Max-Planck-Institut für Meteorologie (Max Planck Institute for Meteorology) | MPI-M | MPI-ESM-MR MPI-ESM-LR MPI-ESM-P |
| Meteorological Research Institute | MRI | MRI-CGCM3 |
| Norwegian Climate Centre | NCC | NorESM1-M NorESM1-ME |

**Table 2. Summary of ocean heat content change. Comparison of CMIP5 models and observations. The median with the 5%-95% confidence interval are presented.**

| Time Period | Depth | CMIP5 linear drift correction ($\times 10^{22}$ J/yr) | CMIP5 quadratic drift correction ($\times 10^{22}$ J/yr) | Observations ($\times 10^{22}$ J/yr) |
|---|---|---|---|---|
| 1970-2005 | 0-700m | 0.42 [0.32, 0.51] | 0.42 [0.32, 0.51] | 0.55[0.50, 0.60] |
| | full depth | 0.69 [0.56, 0.82] | 0.68 [0.54, 0.82] | 0.74 [0.68, 0.80] |
| 1992-2005 | 0-700m | 0.89 [0.77, 1.02] | 0.89 [0.77, 1.02] | 0.85 [0.79, 0.92] |



| | full depth | 1.26 [1.11, 1.42] | 1.25 [1.10, 1.41] | 1.22 [1.16, 1.29] |
| 1970-1991 | 0-700m | 0.39 [0.30, 0.47] | 0.40 [0.31, 0.48] | 0.51 [0.46, 0.56] |
| | full depth | 0.57 [0.44, 0.69] | 0.56 [0.43, 0.68] | 0.61 [0.53, 0.69] |

**Table 3. Summary of total ocean heat content change within 1970-2005 and 1992-2005 by using an alternative method to assess the long-term OHC change. Here the total OHC changes based on observations are calculated by the difference of OHC with 2004-2006 and OHC within 1969-1971 (1990-1992) for 1970-2005 (1992-2005) period to reduce the inter-annual temporal variability. This is an**
5 **alternative method to assess the OHC change in addition to the linear trend in Table 2.**

| Time Period | Depth | CMIP5 linear drift correction ($\times 10^{22}$ J) | CMIP5 quadratic drift correction ($\times 10^{22}$ J) | Observations ($\times 10^{22}$ J) |
| --- | --- | --- | --- | --- |
| 1970-2005 | 0-700m | 16.9 [13.0, 20.0] | 16.7 [12.8, 19.9] | 18.5 [16.5, 20.5] |
| | full depth | 26.6 [22.0, 30.9] | 26.6 [22.2, 31.0] | 28.3 [25.5, 32.3] |
| 1992-2005 | 0-700m | 10.7 [9.0, 12.4] | 10.8 [9.1, 12.5] | 9.0 [8.2, 9.8] |
| | full depth | 15.0 [13.0, 17.1] | 14.9 [12.9, 17.0] | 13.5 [12.3-14.7] |





**Figures**

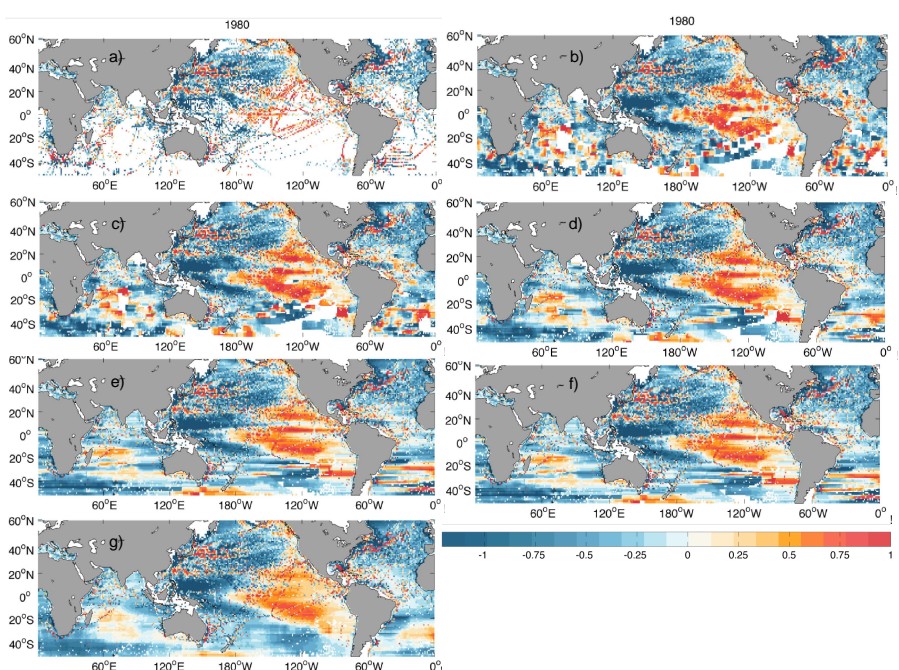

**Figure 1. Gridded Method. a) shows the geographical distribution of OHC700m in 1980 in each 1º by 1º grid, showing good data coverage in Northern Hemisphere and sparse data in Southern Hemisphere. To fill these data gaps by using Gridded Method, OHC in each grid in poorly sampled region (defined as Argo-Ship Area in CZ14) is calculated by averaging OHC in a large latitude-longitude grid with sizes of 5º by 5º, 5º by 10º, 5º by 20º, 1º by 40º, 8º by 40º, and 10º by 40º separately. The resultant OHC distribution is shown from b) to g).**





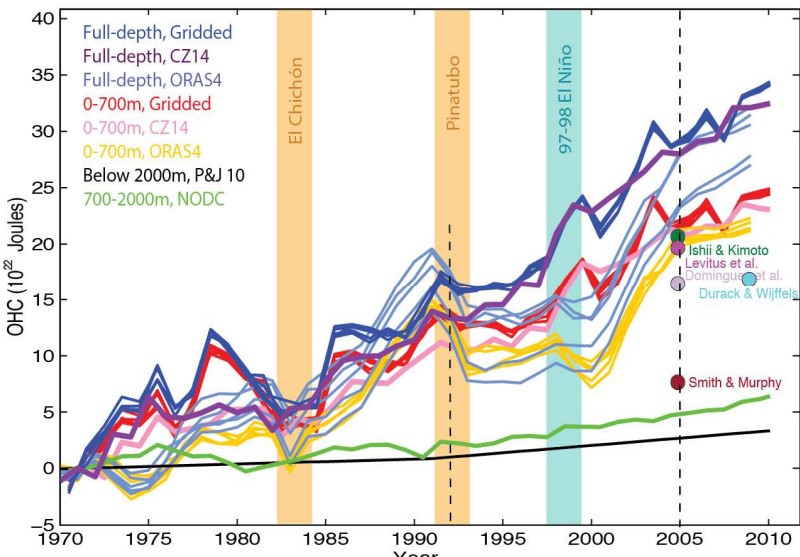

**Figure 2. Observational ocean heat content from 1970 to 2010. 0-700m OHC is shown in red (Gridded method), pink (CZ14 method) and yellow (ORAS4). Five adjusted OHCs presented in *Durack et al.*, (2014) are shown as dots, which are the OHC changes per 35 years. 700-2000m OHC is sourced from NODC in green, and abyssal (2000m - bottom) OHC is from *Purkey and Johnson*, (2010) shown in black (the warming rate within 1970-1991 is scaled to a triple of the linear trend in (*Purkey and Johnson*, 2010)). Full-depth OHC time series are also presented in blue (Gridded method), dark purple (CZ14 method) and light blue (ORAS4). All of the time series are referred to a baseline OHC within the three year period: 1969-1971. The vertical colored bars are 2-years intervals, starting when the event (volcano or El Niño) began.**





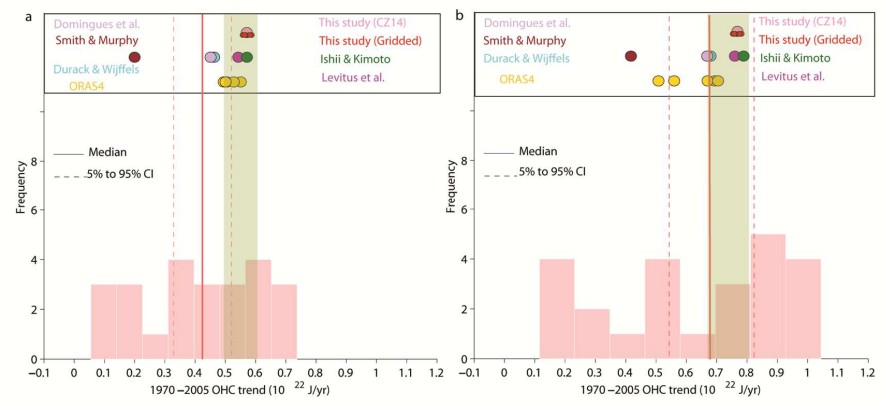

**Figure 3.** OHC trends during 1970-2005 period in observations and CMIP5 models. (a). 0-700m. (b). Full-depth. For models, the histograms are the distribution of CMIP5 results, and the median of the CMIP5 multimodel results is shown in solid line, with 5%-95% confidence interval in dashed lines. For observations, we present the linear trends by different studies: this study (both CZ14 and gridded method), five estimates in *Durack et al.*, (2014) after adjustment, and five ensembles of ORAS4 reanalysis. The 5%-95% confidence intervals for observations are shaded in light green. A quadratic fit to the entire pre-industrial control run was used to correct the CMIP5 time series for model drift.



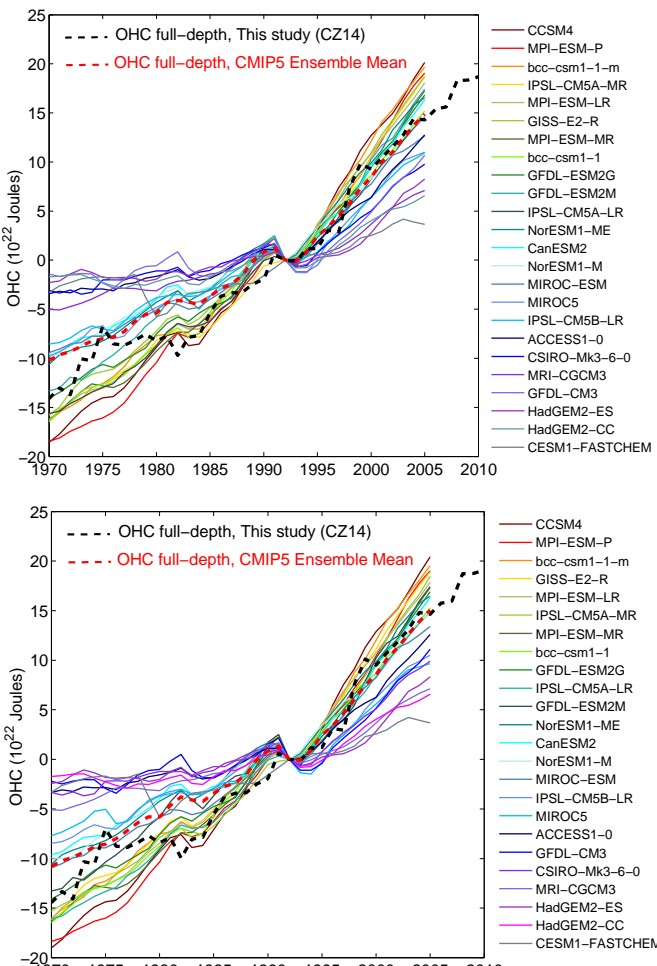

**Figure 4. Full depth OHC by individual CMIP5 models and observations. The observational OHC time series (black dashed) is using CZ14 method (0-700m), *Levitus et al.*, (2012), (700-2000m) and *Purkey and Johnson*, (2010) (2000m-bottom). The multimodel ensemble median is shown in dashed curve. A quadratic fit to the entire pre-industrial control run was used to correct the CMIP5 time series for model drift in the upper panel, and the results for the linear fit are shown in the bottom.**





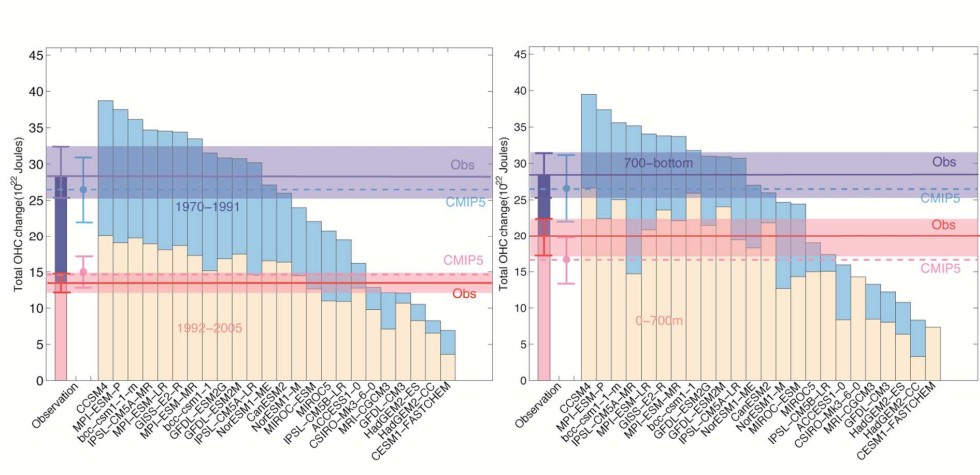

**Figure 5. Comparison of full-depth OHC change between observation and CMIP5 models a). for two separate time periods: 1970-1991 (in blue bars) and 1992-2005 (in red bars) and b). for two vertical layers: 0-700m (in red bars) and 700m-bottom (in blue bars). The medians of the observational total OHC changes are shown in solid lines, compared**
5 **with the model results in dashed lines. Their 5%-95% confidence intervals are presented in error bars. The 5%-95% confidence intervals for observations are also shaded in light red and light blue. A quadratic fit to the entire pre-industrial control run was used to correct the CMIP5 time series for model drift.**