# Peer review of "Observed and simulated full-depth ocean heat content changes for 1970-2005"

_Ocean Science, 2016_

## Short Comment (SC1) · 25 Apr 2016

Overview: In this study, observed and simulated full-depth ocean heat content changes for 1970-2005 has been compared in detail. It is found that the rate of the observed full-depth OHC increasing for 1992-2005 is nearly twice than that for 1970-2005.In addition, the ensemble mean of the CMIP5 models is consistent with the observation. The paper can be published in Ocean Science. Specific comments are listed as below:

1 In the Abstract, the authors noted that "We suggest that OHC be a fundamental metric for climate model validation and evaluation." I think this conclusion can be revised like "We suggest that the ensembles of the CMIP5 models be appropriate for mechanisms for the OHC changes".

2 In the Introduction, the authors stated that "We note that the work presented here is broadly similar to the recent study of Gleckler et al (2016) and provides an important independent verification of some of their key findings."This statement is confused because if the results of this study is broadly similar to the recent study of Gleckler et al (2016), why your studies are different ?

3 In the Data and Methods, the authors used two approaches to map the OHC data. I think these approaches may be not perfect because the ocean state in data-rich areas is not the same to that in data-sparse regions. These approaches should be supplemented with the different ocean dynamic conditions.

4 In the Data and Methods, the authors noted that "Because the upper 700-2000m oceans show an approximate tripling of the heating rate from 1992-2005 compared to 1970-1991 (as shown in Fig. 2, green curve), we assume a proportionate increase in heat uptake in the deep ocean (2000m-bottom)." I think the upper 700-2000m oceans are controlled by the wind while the deep ocean (2000m-bottom) is controlled by the thermohaline circulation. The dynamic conditions are very different. So the assumption is not so appropriate. You can give some observation evidence for this assumption.

5 In the Observation-based full-depth OHC estimates, the authors stated that "The OHC change after the two volcano eruptions is approximately assessed by subtracting the OHC one year before the eruption from the OHC in the second year after

eruption" I think the ocean response to the volcano eruption is not so quick, you can remove the effects of volcano eruptions by longer time delay.

6 In the Observation-based full-depth OHC estimates, the authors noted that "There is also indication of substantial heat discharge from the upper 700m ocean following the extreme 1997-1998 El Niño event". What is the reason of substantial heat discharge from the upper 700m ocean following the extreme 1997-1998 El Niño event?

7 In the Observation-based full-depth OHC estimates, the authors wrote that "Although the comparison between the observational and CMIP5 full-depth OHC results in an insignificant difference, CMIP5 models show a large spread (Figure 3, 4, 5)". What is the dynamics of large spread in these models? I suggest you can explain it in detail.

---

## Short Comment (SC2) · 29 Apr 2016

Overview: In this study, observed and simulated full-depth ocean heat content changes for 1970-2005 has been compared in detail. It is found that the rate of the observed full-depth OHC increasing for 1992-2005 is nearly twice than that for 1970-2005. In addition, the ensemble mean of the CMIP5 models is consistent with the observation.

Reply: Thanks for your evaluation. Yes, we carefully quantify the OHC changes of the improved observational estimates of OHC changes and the drift-corrected CMIP5 model simulations. The model ensemble mean shows a better comparison with the updated observations, but models show large uncertainties.

The paper can be published in Ocean Science. Specific comments are listed as below:

1 In the Abstract, the authors noted that "We suggest that OHC be a fundamental metric for climate model validation and evaluation." I think this conclusion can be revised like "We suggest that the ensembles of the CMIP5 models be appropriate for mechanisms for the OHC changes".

Reply: In this study, we systematically and carefully compare the global OHC changes between observation-based estimates and CMIP5 model simulations for different depth layers and different time periods. We found that the model ensemble mean is consistent with observations but there is large uncertainty in models, which indicates OHC a fundamental metric for model validation. The importance of OHC as a vital climate metric is well-discussed in a recent study of von Schuckmann et al. 2016.

We agree that the CMIP5 models could be very useful for understanding the mechanisms of the OHC changes, but it goes beyond the scope of this study (we do not focus on the mechanisms for OHC changes). We will mention this point in the discussion section in the revised manuscript.

2 In the Introduction, the authors stated that "We note that the work presented here is broadly similar to the recent study of Gleckler et al (2016) and provides an important independent verification of some of their key findings."This statement is confused because if the results of this study is broadly similar to the recent study of Gleckler et al (2016), why your studies are different ?

Reply: Following this statement, we mentioned the differences: "*the present study also makes use of a larger number of CMIP5 models (24 compared to 15) and observation-based estimates of the 0-700m ocean heat content changes (8 compared to 3), including improved XBT bias corrections and new mapping approaches. We are therefore able to more fully characterise the uncertainties associated with CMIP5 models and place our new observation-based estimates of OHC in the context of several previous estimates (including those used by Gleckler et al).*"

3 In the Data and Methods, the authors used two approaches to map the OHC data. I think these approaches may be not perfect because the ocean state in data-rich areas is not the same to that in data-sparse regions. These approaches should be supplemented with the different ocean dynamic conditions.

Reply: We agree that the ocean state in data-rich regions (mainly in Northern Hemisphere) and data-sparse regions (mainly in Southern Hemisphere) are not the same. That is why we argue that some traditional methods those assuming equal OHC change in data-gaps with that in sampled area are probably biased and should be more carefully applied. That is also why we proposed two simple ways to solve this problem: one shown in CZ14 and the other in this study. Also we agree that the two approaches are not perfect, and they are very simple transparent methods as we discussed in the manuscript. There is no "perfect" method since one can never re-observe the ocean in the past. It is very important to develop a more comprehensive method to solve this problem, but this should be provided in another study and the method should be fully evaluated.

4 In the Data and Methods, the authors noted that "Because the upper 700-2000m oceans show an approximate tripling of the heating rate from 1992-2005 compared to 1970-1991 (as shown in Fig. 2, green curve), we assume a proportionate increase in heat uptake in the deep ocean (2000m-bottom)." I think the upper 700-2000m oceans

are controlled by the wind while the deep ocean (2000m-bottom) is controlled by the thermohaline circulation. The dynamic conditions are very different. So the assumption is not so appropriate. You can give some observation evidence for this assumption.

Reply: This is a good point, but the thermohaline circulation also depends on mixing from multiple sources. In fact it is now more commonly referred to as the MOC (meridional overturning circulation) because it is much more than thermal and haline effects, the winds are also very much involved. We added in the manuscript that "the upper 700-2000m oceans are controlled by the wind while the deep ocean (2000m-bottom) is controlled by the thermohaline circulation". The scale we used is a simple/empirical choice. The problem is we don't have sufficient observations below 2000m before 1990, we have to make some assumptions. We discussed the uncertainties induced by this assumption in the manuscript:" *We show that the difference of this lower and upper bound of the 700m-bottom OHC change is equal to ~13% (~10%) of the full-depth OHC change during 1970-1991 (1970-2005), which indicates the maximum error induced by this assumption.*" This size of uncertainty will not impact the key conclusion of this study.

5 In the Observation-based full-depth OHC estimates, the authors stated that "The OHC change after the two volcano eruptions is approximately assessed by subtracting the OHC one year before the eruption from the OHC in the second year after eruption" I think the ocean response to the volcano eruption is not so quick, you can remove the effects of volcano eruptions by longer time delay.

Reply: This is the best choice for assessing the impact of volcanoes. According to Church et al. 2005, there is a strong radiative forcing and heat budget decrease soon after the volcano eruptions (within one year). After two or more years, the natural variabilities (i.e. ENSO) and greenhouse gas forcing comes in and dominate the global EEI, so it experienced a faster recovery.

6 In the Observation-based full-depth OHC estimates, the authors noted that "There is also indication of substantial heat discharge from the upper 700m ocean following the extreme 1997-1998 El Niño event". What is the reason of substantial heat discharge from the upper 700m ocean following the extreme 1997-1998 El Niño event?

Reply: The heat loss by the global ocean after El Nino has been well-documented by studies such as Trenberth et al. 2002 and Balmaseda et al. 2013, I quote a discussion in Balmaseda et al. 2013 here." *there is an additional cooling episode following the huge 1997– 1998 El Niño event after 1998, which mainly affects the upper 700 m. The event led to a global warming of the atmosphere and made 1998 the warmest year on record to that point as heat came out of the ocean, largely through evaporative cooling [Trenberth et al., 2002].*"

7 In the Observation-based full-depth OHC estimates, the authors wrote that "Although the comparison between the observational and CMIP5 full-depth OHC results in an insignificant difference, CMIP5 models show a large spread (Figure 3, 4, 5)". What is the dynamics of large spread in these models? I suggest you can explain it in detail.

Reply: This is a good question, and this is definitely a hot topic in climate community recently, but it is not solved yet. Unfortunately, some of this is simply that some models are not very good, they may not conserve properly, their physical parameterizations are not state of the art, they do not replicate the past very well, and so forth. We just provided several possible reasons according to the recent literatures:" *The reasons why the models have large divergence are still an actively discussed issue. Frolicher et al., (2015) discussed the large range of model results and attributed a contribution of this to the differences in indirect aerosols. Additionally, CMIP5 has been missing post 2000 volcanic eruptions in these simulations as discussed in Glecker et al (2016), but this effect is shown to small and less than 0.1 W m$^{-2}$ as indicated in Trenberth et al., (2014).*".

---

## Referee Comment (RC1) · Anonymous Referee #1 · 2 May 2016

Review of the Manuscript "Observed and simulated full-depth ocean heat content changes for 1970-2005" by Lijing Cheng et al.

This is an interesting paper, which along with other recently published and in-press publications stress the importance of the ocean heat content calculations for the climate monitoring and provides an update of both the observation- and model-based ocean heat content estimates.

General comments

1) I believe the paper should more strongly underline the conclusion that the existing (CMIP5) models are characterized by the extremely large spread in their estimates of the total OHC. This spread exceeds by far the estimated observational uncertainty in the OHC change even for the upper 0-700m where the the model drift is expected to

be less important compared to the deeper layer. This finding poses the question about the ability of the existing models to simulate the OHC change.

2)There is a small discussion in the manuscript regarding the OHC decrease after the major volcano eruptions (page 6). The OHC time series shows several other OHC-decrease events of similar magnitude (1976 2001, 2004, 2007) which are obviously not connected to any big volcanic eruption. Consequently, the interpretation of the OHC-decrease events near 1983 and 1993 as being most probably due to the volcanic eruptions allows alternative explanations as well.

Minor comments

P.1 Lines 25ff: "Numerous efforts have been done to detect the historical OHC change" . The list of references which follows in parenthesis is incomplete: for instance the earlier estimates done by the Levitus group are missing, as well as the estimates provided by Gouretski&Koltermann, 2007. The reader thus gets a wrong impression that the cited studies are the only available in the literature where the OHC estimates have been reported. Please, extend the list of relevant studies. P.2, lines 16ff Please, reformulate the sentence: for instance, the following piece of text "to construct the climatology based on data with near-global data coverage" definitely needs improvement.

P.2, line 30: "The gridded method". Though this term has already been used by the authors in the earlier paper, I suggest to change the name to, say, "multiple grid method", or "flexible grid method", because otherwise the name of the metod states that the method itself is subjected to gridding, whereas it is the data what is gridded.

Data and Method section: there is no mention here about the usage (or not usage) of the Mechanical bathythermograph data. This is a large data set, with the data being biased. Gouretski&Reseghetti 2010 provided a correction scheme for the MBT data, which successfully reduces the overall bias.

Page 4, line 3: change to "less land and no boundary currents"

Page 4, line 14ff: "The five ensemble members ... sample plausible uncertainties" - 1) how the uncertainties can be sampled and 2) what is the method to decide if they are plausible???

Page 4 line 21: change "from choice" to "from the choice"

Page 4, line 29ff: Is the assumption about the proportional increase in heat uptake in the deep ocean justified?

Page 5, line 9: "i.g. spurious long-term trends arising the slow model adjustment...." - are the words "due to" missing here??

Page 6 line 10: "The total OHC change ... has increased ..." - please, indicate the time period here.

Page 9, line 24: change "is shown to small" to "is shown to be small"

---

## Referee Comment (RC2) · M. Balmaseda (Referee) · 9 May 2016

The manuscript presents a new estimates of global ocean heat content temporal evolution for the period 1970-2005, and compare them with previous observational estimates. The authors then use an ensemble of observational estimates to evaluate the OHC trends in an ensemble of CMIP5 model integrations. They find that the median of the CMIP5 ensemble agrees well with the observational estimates of OHC global trends, both of them showing an acceleration of ocean warming during the period 1992-2005. They and propose to use OHC as a metric to evaluate climate models. The paper is clear and well written: the problem in question is well introduced, the results clearly presented, and there is a levelled correspondence between the numerical findings and the interpretation given.

[Figure]

I have some questions and comments that the authors may want to take into account.

• Abstract: "We suggest that OHC be a fundamental metric for climate model validation and evaluation". The current study only deals with trends of global OHC for a given period of time. Maybe this should be the specific metric proposed. Otherwise the current statement in the last sentence of the abstract is far too generic, and open to miss-interpretation.

• Why the validation period does not extend beyond 2005? The period post-2005, when the so-call hiatus started, is of large interest. Can the authors comment on their choice of period? Would the choice of period change their conclusions?

• In the observational estimates, the corrections by Durack etal (2014) seem to be included in some of the ensembles. Those use CMIP5 model information to fill the gaps. Then, these corrected estimates are used to validate CMIP5 models. It seems to me like a circular argument. How would the results be influenced by removing the Durack etal (2014) corrections?

• If the median is chosen against the mean in recognition of the non-gaussianity of the distribution, the use of Gaussian estimations for the confidence levels (twice the standard deviations) to evaluate the significance of the median seems inconsistent. Are there any other ways of estimating confidence levels for the median using non-parametric distribution?

• It is said in the text that the estimate of OHC by Smith and Murphy is discounted because the values are smaller than the others. This is quite an adhoc reason. Can the authors provide a more solid motivation for excluding the estimation?. The estimate is not removed from figure 3, which is misleading

---

## Short Comment (SC3) · 12 May 2016

Please find our responses and the revised manuscript in the Supplement zip file.

Please also note the supplement to this comment:
http://www.ocean-sci-discuss.net/os-2016-16/os-2016-16-SC3-supplement.zip

———————————————

---

## Short Comment (SC4) · 12 May 2016

Dr. M. Balmaseda, Please find our responses and the revised manuscript in the Supplement zip file.

Please also note the supplement to this comment:
http://www.ocean-sci-discuss.net/os-2016-16/os-2016-16-SC4-supplement.zip
* * *

---

## Editor Comment (EC1) · M. Hecht (Editor) · 13 May 2016

Dear Dr. Chen and co-authors,

with the "discussion period" for your manuscript scheduled to remain open until June 3, we will allow for further comments until that date. At that point, if no other comments have been received, you may wish to simply upload the same revised version of your manuscript, but with the link that will be provided at that point.

I look forward to seeing your paper through the review process.

Sincerely Yours, –Matthew Hecht
* * *

---

## Author Comment (AC1) · 6 Jun 2016

We are grateful to the two reviewers and the community commenter.

Please also note the supplement to this comment:
http://www.ocean-sci-discuss.net/os-2016-16/os-2016-16-AC1-supplement.zip

---

## Author Response (AR1)

We thank the anonymous referee#1 for the comments on our manuscript submitted to *ocean science*. We appreciate the thoughtful and constructive feedback on the paper. We have addressed all concerns in the revised manuscript, as documented below in our point-by-point responses (in blue) to reviewer comments (in black)

Review of the Manuscript "Observed and simulated full-depth ocean heat content changes for 1970-2005" by Lijing Cheng et al.

This is an interesting paper, which along with other recently published and in-press publications stress the importance of the ocean heat content calculations for the climate monitoring and provides an update of both the observation- and model-based ocean heat content estimates.

General comments

1) I believe the paper should more strongly underline the conclusion that the existing (CMIP5) models are characterized by the extremely large spread in their estimates of the total OHC. This spread exceeds by far the estimated observational uncertainty in the OHC change even for the upper 0-700m where the model drift is expected to be less important compared to the deeper layer. This finding poses the question about the ability of the existing models to simulate the OHC change.

Reply: We agree that the paper should more strongly underline the large spread of CMIP5 models. Therefore, we made the following revisions:

(1). In the abstract, a sentence is added to highlight this issue:"

*The CMIP5 models show large spread in OHC changes, suggesting that some models are not state-of-the-art and require further improvements. However, the ensemble median has excellent agreement with our observational estimate*"

(2). In Page10Lines1-3. This paragraph is to highlight the large uncertainty of climate models. One more sentence is added "*The spread of CMIP5 models far exceeds the estimated observational uncertainty in the OHC changes even for the upper 0-700m where the model drift is expected to be less important compared to the deeper layer.*"

We think it is better not to "**poses the question about the ability of the existing models to simulate the OHC change**" as suggested by Reviewer#1. Because some of the models are not good according to our assessments but some others are consistent with observations, so a more precise conclusion is preferred.

2) There is a small discussion in the manuscript regarding the OHC decrease after the major volcano eruptions (page 6). The OHC time series shows several other OHC decrease events of similar magnitude (1976 2001, 2004, 2007) which are obviously not connected to any big volcanic eruption. Consequently, the interpretation of the OHC-decrease events near 1983 and 1993 as being most probably due to the volcanic eruptions allows alternative explanations as well.

Reply: This is a good point. We can't rule out the possibility that the OHC-decrease events in 1983/1993 are due to other forces, based on the observation-based analyses because all of the signals are mixed together.

A discussion is added in Page6Line30-Page7Line6:" *But our observational analyses can not exclude the possibility that the unforced ocean variability (such as ENSO) and the insufficiency of data coverage (which could induce spurious inter-annual OHC change) are fully or partly responsible for the values calculated above, which requires more careful model-based studies in the future. Moreover, it is also suggested that volcanic eruptions can trigger an El Niño like response in the ocean, which is another possible explanation (Mann et al., 2005).*"

Minor comments

P.1 Lines 25ff: "Numerous efforts have been done to detect the historical OHC change". The list of references which follows in parenthesis is incomplete: for instance the earlier estimates done by the Levitus group are missing, as well as the estimates provided by Gouretski&Koltermann, 2007. The reader thus gets a wrong impression that the cited studies are the only available in the literature where the OHC estimates have been reported. Please, extend the list of relevant studies.

Reply: We added two more references in the revised manuscript (*Levitus et al.*, 2005; *Gouretski and Koltermann*, 2007), and added "for example" to note that we only listed parts of but the latest references (since it is not a review paper).

P.2, lines 16ff. Please, reformulate the sentence: for instance, the following piece of text "to construct the climatology based on data with near-global data coverage" definitely needs improvement.

Reply: This sentence is modified to "*Because the choice of reference climatology to compute anomalies can lead to errors due to the sparseness and inhomogeneity of the historical ocean sampling (Lyman and Johnson, 2014; Cheng and Zhu, 2015), it is preferable to use the climatology which is constructed based on data with near-global data coverage (Cheng and Zhu, 2015), i.e., during the recent years in the Argo period.*"

P.2, line 30: "The gridded method". Though this term has already been used by the authors in the earlier paper, I suggest to change the name to, say, "multiple grid method", or "flexible grid method", because otherwise the name of the metod states that the method itself is subjected to gridding, whereas it is the data what is gridded.

Reply: Thanks for the comment. The gridded method is changed to "Flexible-grid Method" in the revised manuscript.

Data and Method section: there is no mention here about the usage (or not usage) of the Mechanical bathythermograph data. This is a large data set, with the data being biased. Gouretski&Reseghetti 2010 provided a correction scheme for the MBT data, which successfully reduces the overall bias.

Reply: We have corrected the MBT bias by using the method provided in *Ishii and Kimoto*, (2009). We mentioned this point in Data and Method section (Page3Line19):" *MBT bias is corrected using the method provided in Ishii and Kimoto, (2009).*".

Page 4, line 3: change to "less land and no boundary currents"

Reply: Done

Page 4, line 14ff: "The five ensemble members ... sample plausible uncertainties" - 1) how the uncertainties can be sampled and 2) what is the method to decide if they are plausible???

Reply: This sentence might be misleading. Now it is revised to "*The five ensemble members in ORAS4 approximately represent the uncertainties in the wind forcing, observation coverage, and the deep ocean.*"

Page 4 line 21: change "from choice" to "from the choice"

Reply: Done

Page 4, line 29ff: Is the assumption about the proportional increase in heat uptake in the deep ocean justified?

Reply: It is very difficult to provide a full justification for this assumption. Because (1). There is no sufficient observations pre-1990 below 2000m (2). The models have large discrepancies about the deep ocean changes below 2000m. So how much the deep ocean changed is unknown and it is difficult to have a direct estimate. We simply provided two justifications: (1) The upper 7000m oceans are mostly

controlled by the wind, while the deep oceans (700m-bottom) are mostly controlled by the meridional overturning circulation. So the OHC changes at 700-2000m and 2000-bottom may share some similarities. (2) And we provided the uncertainties involved, suggesting that this assumption will not significantly impact our key conclusion." *We show that the difference of this lower and upper bound of the 700m-bottom OHC change is equal to ~13% (~10%) of the full-depth OHC change during 1970-1991 (1970-2005), which indicates the maximum error induced by this assumption.*"

Page 5, line 9: "i.g. spurious long-term trends arising the slow model adjustment...." - are the words "due to" missing here??

Reply: Modified to "*spurious long-term trends arising due to the slow model adjustment to the initial conditions and/or imperfect representation of the energy budget*"

Page 6 line 10: "The total OHC change ... has increased ..." - please, indicate the time period here.

Reply: The time period "*from 1970 to 2005*" is indicated here.

Page 9, line 24: change "is shown to small" to "is shown to be small"

Reply: Done

We thank the referee#2 (Dr. M. Balmaseda) for the comments on our manuscript submitted to *ocean science*. We appreciate the thoughtful and constructive feedback on the paper. We have addressed all concerns in the revised manuscript, as documented below in our point-by-point responses (in blue) to the comments (in black)

The manuscript presents a new estimates of global ocean heat content temporal evolution for the period 1970-2005, and compare them with previous observational estimates. The authors then use an ensemble of observational estimates to evaluate the OHC trends in an ensemble of CMIP5 model integrations. They find that the median of the CMIP5 ensemble agrees well with the observational estimates of OHC global trends, both of them showing an acceleration of ocean warming during the period 1992-2005. They and propose to use OHC as a metric to evaluate climate models. The paper is clear and well written: the problem in question is well introduced, the results clearly presented, and there is a levelled correspondence between the numerical findings and the interpretation given.

I have some questions and comments that the authors may want to take into account

1. Abstract: "We suggest that OHC be a fundamental metric for climate model validation and evaluation". The current study only deals with trends of global OHC for a given period of time. Maybe this should be the specific metric proposed. Otherwise the current statement in the last sentence of the abstract is far too generic, and open to miss-interpretation.

Reply: Yes it is a good point. We modified the last sentence in the abstract to "*We suggest that OHC be a fundamental metric for climate model validation and evaluation especially for forced changes (decadal time-scales)*"

2. Why the validation period does not extend beyond 2005? The period post-2005, when the so-call hiatus started, is of large interest. Can the authors comment on their choice of period? Would the choice of period change their conclusions?

Reply: There are several reasons not extending the period beyond 2005 in the current study: (1). Observational-based OHCs from Durack et al 2005 end at 2005. (2). CMIP5 historical runs end at 2005. Post-2005 runs are projections, which are not correctly forced by the climate forcings: such as greenhouse gas emissions and volcanic eruptions. Also there is often a discontinuity in observational estimates of ocean heat content due to transition from mostly XBT measurements to mostly Argo measurements

3. In the observational estimates, the corrections by Durack etal (2014) seem to be included in some of the ensembles. Those use CMIP5 model information to fill the gaps. Then, these corrected estimates are used to validate CMIP5 models. It seems to me like a circular argument. How would the results be influenced by removing the Durack etal (2014) corrections?

Reply: It is a good point. We have now tested this. We have explicitly discussed this point in the main context (page8 line19-25)

"*Because the Durack et al. (2014) global OHC adjustments are partly based on heat uptake in the CMIP5 models, they should not be used to then evaluate the models. When removing Durack et al. (2014) estimates, the median change within 1970-2005 is 0.56 $\times 10_{22}$ J/yr for OHC0-700m and 0.75 $\times 10_{22}$ J/yr for OHC0-700m, both of which are nearly identical with the results in Table 2, suggesting that including Durack et al. (2014) does not influence the main conclusion of our study.*".

4. If the median is chosen against the mean in recognition of the non-gaussianity of the distribution, the use of Gaussian estimations for the confidence levels (twice the standard deviations) to evaluate the significance of the median seems inconsistent. Are there any other ways of estimating confidence levels for the median using nonparametric distribution?

Reply: To calculate the model ensemble results, we used median to be consistent with the observation-based results. This is to remove the impact of outliers, since the sample size is small for both observation and CMIP5 based OHC estimates. For the confidence intervals, there is no a priori reason for the statistics to be non-Gaussian other than there is a small sample and the likelihood that there are some outliers. For model results, we test the difference of the results by using the following three strategies: (1) Median (used in this study); (2) Mean; and (3) Mean after removing the minimum and the maximum of the model results. The third method is to estimate the percentiles by ranking the model trends and reading off the percentiles (E.g. if there are 24 models, the $10_{th}$ percentile is the $2.4_{th}$ model, so we remove the minimum and the maximum of the models to get $5_{th}$-$95_{th}$ percentiles).

**OHC0-700m**
1970-2005: Median: $0.42\times10_{22}$ J/yr
1970-2005: Mean: $0.42\times10_{22}$ J/yr
1970-2005: Mean (remove the minimum and maximum): $0.42\times10_{22}$ J/yr
**OHC full-depth**
1970-2005: Median: $0.68\times10_{22}$ J/yr
1970-2005: Mean: $0.66\times10_{22}$ J/yr
1970-2005: Mean (remove the minimum and maximum): $0.66\times10_{22}$ J/yr
This test indicates that it makes no much change on the results when using the three different strategies.

Based on these discussions, we still decided to use the standard deviation to characterize the spread and the median to characterize the ensemble average – essentially because we do not have enough models to have a good statistics. We found it could be helpful to include a discussion with respect to this point in section 3, see page13line17:

"*Furthermore, the OHC for models show a non-Gaussian distribution (Figure 3), potentially challenging our method of the use of Gaussian estimations for the confidence levels. However, there is no a priori reason for the statistics to be non-Gaussian other than there is a small sample and the likelihood that there are some outliers. The non-Gaussian nature of the distribution (Fig.3) may be partly due to the small sample size. The use of the median reduces the impact of outliers and then enables us to use the standard deviation to characterize the spread.*"

5. It is said in the text that the estimate of OHC by Smith and Murphy is discounted because the values are smaller than the others. This is quite an adhoc reason. Can the authors provide a more solid motivation for excluding the estimation?. The estimate is not removed from figure 3, which is misleading

Reply: (1). Fully evaluating an OHC estimate (such as Smith&Murphy2007) requires a more comprehensive study in the future (to understand the mapping methods etc.). Our decision to remove Smith&Murphy is based on fig.2, because apparently it is an outlier. Why this is so is beyond the scope of this study.
(2). We still keep Smith&Murphy2007 in Fig.3. And we note in the main context that including Smith&Murphy2007 value does not impact our results, since we use median rather than mean (Page6line11).

We thank "xiao boot" for the comments on our manuscript submitted to *ocean science*. Here we provide our responses in blue.

Overview: In this study, observed and simulated full-depth ocean heat content changes for 1970-2005 has been compared in detail. It is found that the rate of the observed full-depth OHC increasing for 1992-2005 is nearly twice than that for 1970-2005. In addition, the ensemble mean of the CMIP5 models is consistent with the observation.

Reply: Thanks for your evaluation. Yes, we carefully quantify the OHC changes of the improved observational estimates of OHC changes and the drift-corrected CMIP5 model simulations. The model ensemble mean shows a better comparison with the updated observations, but models show large uncertainties.

The paper can be published in Ocean Science. Specific comments are listed as below:

1 In the Abstract, the authors noted that "We suggest that OHC be a fundamental metric for climate model validation and evaluation." I think this conclusion can be revised like "We suggest that the ensembles of the CMIP5 models be appropriate for mechanisms for the OHC changes".

Reply: In this study, we systematically and carefully compare the global OHC changes between observation-based estimates and CMIP5 model simulations for different depth layers and different time periods. We found that the model ensemble mean is consistent with observations but there is large uncertainty in models, which indicates OHC a fundamental metric for model validation. The importance of OHC as a vital climate metric is well-discussed in a recent study of von Schuckmann et al. 2016.

We agree that the CMIP5 models could be very useful for understanding the mechanisms of the OHC changes, but it goes beyond the scope of this study (we do not focus on the mechanisms for OHC changes). We will mention this point in the discussion section in the revised manuscript.

2 In the Introduction, the authors stated that "We note that the work presented here is broadly similar to the recent study of Gleckler et al (2016) and provides an important independent verification of some of their key findings."This statement is confused because if the results of this study is broadly similar to the recent study of Gleckler et al (2016), why your studies are different ?

Reply: Following this statement, we mentioned the differences: "*the present study also makes use of a larger number of CMIP5 models (24 compared to 15) and observation-based estimates of the 0-700m ocean heat content changes (8 compared to 3), including improved XBT bias corrections and new mapping approaches. We are therefore able to more fully characterise the uncertainties associated with CMIP5 models and place our new observation-based estimates of OHC in the context of several previous estimates (including those used by Gleckler et al).*"

3 In the Data and Methods, the authors used two approaches to map the OHC data. I think these approaches may be not perfect because the ocean state in data-rich areas is not the same to that in data-sparse regions. These approaches should be supplemented with the different ocean dynamic conditions.

Reply: We agree that the ocean state in data-rich regions (mainly in Northern Hemisphere) and data-sparse regions (mainly in Southern Hemisphere) are not the same. That is why we argue that some traditional methods those assuming equal OHC change in data-gaps with that in sampled area are probably biased and should be more carefully applied. That is also why we proposed two simple ways to solve this problem: one shown in CZ14 and the other in this study. Also we agree that the two approaches are not perfect, and they are very simple transparent methods as we discussed in the manuscript. There is no "perfect" method since one can never re-observe the ocean in the past. It is very important to develop a more comprehensive method to solve this problem, but this should be provided in another study and the method should be fully evaluated.

4 In the Data and Methods, the authors noted that "Because the upper 700-2000m oceans show an approximate tripling of the heating rate from 1992-2005 compared to 1970-1991 (as shown in Fig. 2, green curve), we assume a proportionate increase in heat uptake in the deep ocean (2000m-bottom)." I think the upper 700-2000m oceans are controlled by the wind while the deep ocean (2000m-bottom) is

controlled by the thermohaline circulation. The dynamic conditions are very different. So the assumption is not so appropriate. You can give some observation evidence for this assumption.

Reply: This is a good point, but the thermohaline circulation also depends on mixing from multiple sources. In fact it is now more commonly referred to as the MOC (meridional overturning circulation) because it is much more than thermal and haline effects, the winds are also very much involved. We added in the manuscript that "the upper 700-2000m oceans are controlled by the wind while the deep ocean (2000m-bottom) is controlled by the thermohaline circulation". The scale we used is a simple/empirical choice. The problem is we don't have sufficient observations below 2000m before 1990, we have to make some assumptions. We discussed the uncertainties induced by this assumption in the manuscript:" *We show that the difference of this lower and upper bound of the 700m-bottom OHC change is equal to ~13% (~10%) of the full-depth OHC change during 1970-1991 (1970-2005), which indicates the maximum error induced by this assumption.*" This size of uncertainty will not impact the key conclusion of this study.

5 In the Observation-based full-depth OHC estimates, the authors stated that "The OHC change after the two volcano eruptions is approximately assessed by subtracting the OHC one year before the eruption from the OHC in the second year after eruption" I think the ocean response to the volcano eruption is not so quick, you can remove the effects of volcano eruptions by longer time delay.

Reply: This is the best choice for assessing the impact of volcanoes. According to Church et al. 2005, there is a strong radiative forcing and heat budget decrease soon after the volcano eruptions (within one year). After two or more years, the natural variabilities (i.e. ENSO) and greenhouse gas forcing comes in and dominate the global EEI, so it experienced a faster recovery.

6 In the Observation-based full-depth OHC estimates, the authors noted that "There is also indication of substantial heat discharge from the upper 700m ocean following the extreme 1997-1998 El Niño event". What is the reason of substantial heat discharge from the upper 700m ocean following the extreme 1997-1998 El Niño event?

Reply: The heat loss by the global ocean after El Nino has been well-documented by studies such as Trenberth et al. 2002 and Balmaseda et al. 2013, I quote a discussion in Balmaseda et al. 2013 here." *there is an additional cooling episode following the huge 1997– 1998 El Niño event after 1998, which mainly affects the upper 700 m. The event led to a global warming of the atmosphere and made 1998 the warmest year on record to that point as heat came out of the ocean, largely through evaporative cooling [Trenberth et al., 2002].*"

7 In the Observation-based full-depth OHC estimates, the authors wrote that "Although the comparison between the observational and CMIP5 full-depth OHC results in an insignificant difference, CMIP5 models show a large spread (Figure 3, 4, 5)". What is the dynamics of large spread in these models? I suggest you can explain it in detail.

Reply: This is a good question, and this is definitely a hot topic in climate community recently, but it is not solved yet. Unfortunately, some of this is simply that some models are not very good, they may not conserve properly, their physical parameterizations are not state of the art, they do not replicate the past

very well, and so forth.  We just provided several possible reasons according to the recent literatures:"

*The reasons why the models have large divergence are still an actively discussed issue. Frolicher et al., (2015) discussed the large range of model results and attributed a contribution of this to the differences in indirect aerosols. Additionally, CMIP5 has been missing post 2000 volcanic eruptions in these simulations as discussed in Glecker et al (2016), but this effect is shown to small and less than 0.1 W m$^{-2}$ as indicated in Trenberth et al., (2014).".*